

# 60S dynamic state of bacterial ribosome is fixed by yeast mitochondrial initiation factor 3

Sergey Levitskii[1], Ksenia Derbikova[1], Maria V. Baleva[1], Anton Kuzmenko[1,2], Andrey V. Golovin[3,4,5], Ivan Chicherin[1], Igor A. Krasheninnikov[1] and Piotr Kamenski[1]

[1] Faculty of Biology, Moscow State University, Moscow, Russia
[2] Current affiliation: Institute of Molecular Genetics, Russian Academy of Sciences, Moscow, Russia
[3] Faculty of Bioengineering and Bioinformatics, Moscow State University, Moscow, Russia
[4] Institute of Molecular Medicine, Sechenov First Moscow State Medical University, Moscow, Russia
[5] Faculty of Computer Science, Higher School of Economics, Moscow, Russia

Corresponding author
Piotr Kamenski,
peter@protein.bio.msu.ru

## ABSTRACT

The processes of association and dissociation of ribosomal subunits are of great importance for the protein biosynthesis. The mechanistic details of these processes, however, are not well known. In bacteria, upon translation termination, the ribosome dissociates into subunits which is necessary for its further involvement into new initiation step. The dissociated state of the ribosome is maintained by initiation factor 3 (IF3) which binds to free small subunits and prevents their premature association with large subunits. In this work, we have exchanged IF3 in *Escherichia coli* cells by its ortholog from *Saccharomyces cerevisiae* mitochondria (Aim23p) and showed that yeast protein cannot functionally substitute the bacterial one and is even slightly toxic for bacterial cells. Our in vitro experiments have demonstrated that Aim23p does not split *E. coli* ribosomes into subunits. Instead, it fixes a state of ribosomes characterized by sedimentation coefficient about 60S which is not a stable structure but rather reflects a shift of dynamic equilibrium between associated and dissociated states of the ribosome. Mitochondria-specific terminal extensions of Aim23p are necessary for "60S state" formation, and molecular modeling results point out that these extensions might stabilize the position of the protein on the bacterial ribosome.

## INTRODUCTION

Upon termination of protein biosynthesis in bacteria, 70S ribosome dissociates into small (30S) and large (50S) subunits. Free small subunit then takes part in de novo formation of the initiation complex with mRNA, initiator tRNA, and several initiation factors. Binding of the large subunit promotes release of the initiation factors, and assembled 70S ribosomes begins a new round of translation (for review, see *Laursen et al., 2005*).

It is known that two proteins, namely RRF and EF-G, are responsible for bacterial ribosomes dissociation into subunits after termination of protein biosynthesis

PeerJ _______________________________________________________

(*Zavialov, Hauryliuk & Ehrenberg, 2005*; *Peske, Rodnina & Wintermeyer, 2005*). Once free 30S and 50S subunits appear, initiation factor 3 (IF3) binds the small subunit in order to keep it apart from the large one (*Zavialov, Hauryliuk & Ehrenberg, 2005*). This stage is, in fact, the very first stage of the translation initiation process; 30S•IF3 complex becomes the basis for the full-size initiatory complex formation which includes Shine-Dalgarno sequence of mRNA, initiator tRNA, and initiation factors 1 and 2. It is worth mentioning that anti-association activity of IF3 is definitely of passive mode: it does not promote dissociation of the ribosome into subunits but instead binds to free small subunit and prevents its re-association with the large one (*Gualerzi, Risuleo & Pon, 1977*; *Gottleib, Davis & Thompson, 1975*).

The exact mechanism of ribosome dissociation into subunits remains unclear. This is due to methodological complications of studying this fast and dynamic process. In kinetic study, a model was proposed that assumed the existence of several consecutive conformations of ribosome in course of its dissociation; IF3 was hypothesized to be a potential effector of corresponding conformational changes which could shift the equilibria between different states of dissociating ribosome (*Goss, Parkhurst & Wahba, 1980*). It can be assumed that these conformations might be characterized by different sedimentation coefficients, less than 70S but probably more than 50S. Indeed, a ribosomal ~60S state was described in in vitro experiments; this state was formed under specific experimental conditions (*Morimoto, 1969*). The authors used a term "60S component" and postulated that this was a new stable intermediate of the subunits dissociation/association reaction and that this intermediate was just "swollen 70S" (*Morimoto, 1969*). Further investigations, however, have demonstrated that the exact sedimentation coefficient of this "swollen 70S" depends on the sedimentation speed and on the initial 70S concentration (*Spirin, 1971*). These results fit very well to the above-discussed hypothesis about consecutive conformational changes of 70S ribosome during dissociation. However, none of these intermediate states has been seen as stable structure.

In this work, we investigated the effects of yeast mitochondrial IF3, Aim23p, on the *Escherichia coli* translation. The idea has come from the recent work of *Ayyub et al. (2018)* where it has been demonstrated that mammalian mitochondrial IF3 (mtIF3), although being unable to fully substitute for IF3 in *E. coli*, exhibits some functional activity in bacterial cells. We exchanged the *infC* gene (coding for IF3) in bacteria by the AIM23 gene and found that Aim23p cannot substitute the cognate factor. Moreover, Aim23p was slightly toxic for the bacterial cell which was mediated by mitochondria-specific parts of the protein, namely its N- and C-terminal extensions. Our in vitro investigations have revealed that Aim23p does bind to *E. coli* ribosome and fixes its unusual state with sedimentation coefficient about 60S. This state can be further transformed into fully dissociated state if Am23p concentration is increased. According to the results of wet experiments and molecular modeling, terminal extensions of Aim23p might be responsible for 60S state fixation.

## MATERIALS AND METHODS

**Plasmids**, *E. coli* **strains**, and **oligonucleotides** used in the work may be found in Tables 1, 2 and 3, respectively.

**Table 1 Plasmids used in this work.**

| Plasmid | Description |
|---|---|
| **pACDH** | Low-copy vector for expression in *E. coli* |
| pACDHinfC* | pACDH with cloned *infC* gene from *E. coli* |
| pACDHAim23* | pACDH with cloned AIM23 gene lacking mitochondrial targeting sequence |
| pACDHAim23ΔNΔC* | pACDH with cloned AIM23 gene lacking mitochondrial targeting sequence and both terminal extensions |
| **pBAD** | Vector for *E. coli* expression containing glucose-repressible promoter |
| pBADinfC* | pBAD with cloned *infC* gene from *E. coli* |
| **pKD3** | Plasmid containing FRT-cat-FRT for preparation of *E. coli* disruption cassettes |
| **pKD46** | Plasmid with Lambda Red recombinase from phage λ for efficient gene disruption in *E. coli* |
| **pET32a** | Vector for the heterologous proteins expression in *E. coli* |
| pETIF3* | pET32a with cloned *infC* gene from *E. coli* |
| pETAim23* | pET32a with cloned AIM23 gene lacking mitochondrial targeting sequence |
| pETAim23ΔNΔC* | pET32a with cloned AIM23 gene lacking mitochondrial targeting sequence and both terminal extensions |

Note:
* Generated in this work.

**Table 2 *E. coli* strains used in this work.**

| Strain | Genotype/description/purpose |
|---|---|
| **MG 1655** | K-12 F⁻ λ⁻ilvG⁻ rfb-50 rph-1 |
| | For genetic manipulations, for ribosome isolation |
| MG_infC_ACDH* | MG 1655 + pACDHinfC + pKD46 |
| MG_ΔIF3* | MG_infC_ACDH with first 153 nucleotides of *inf3* gene exchanged by chloramphenicol resistance cassette |
| MG_infC_BAD* ("vector" in Fig. 3) | MG 1655 + pBADinfC + pACDH |
| MG_IF3* ("IF3" in Fig. 3) | MG 1655 + pBADinfC + pACDHinfC |
| MG_Aim23* ("Aim23" in Fig. 3) | MG 1655 + pBADinfC + pACDHAim23 |
| MG_Aim23ΔNΔC* ("Aim23ΔNΔC" in Fig. 3) | MG 1655 + pBADinfC + pACDHAim23ΔNΔC |
| **Rosetta (DE3) pLysS** | F⁻ ompT hsdSB(RB⁻ mB⁻) gal dcm λ(DE3 [lacI lacUV5-T7 gene 1 ind1 sam7 nin5]) pLysSRARE (CamR) |
| | For heterologous proteins synthesis and purification |
| **Top10** | F⁻ mcrA Δ(mrr-hsdRMS-mcrBC) φ80lacZΔM15 ΔlacX74 nupG recA1 araD139 Δ(ara-leu)7697 galE15 galK16 rpsL(StrR) endA1 λ⁻ |
| | For molecular cloning |

Note:
* Generated in this work.

## Cloning and standard procedures

Different versions of AIM23 (*Saccharomyces cerevisiae*) and *infC* (*E. coli*) genes were cloned into above-mentioned vectors by standard polymerase chain reaction (PCR)-restriction-ligation approach.

**Table 3 Oligonucleotides used in this work.**

| | | |
|---|---|---|
| 1 | Cloning of IF3 into pACDH and pBAD | tcagccatggctaaaggcggaaaacgagttc |
| 2 | | tcaggaattcctactgtttcttcttaggagcga |
| 3 | Cloning of AIM23ΔNΔC into pACDH | tcagccatggcttggagcaccgggaca |
| 4 | | tcaggaattcctatggtttaacgtcctttggta |
| 5 | Cloning of AIM23 into pACDH | tcagccatggctaatgcatcatctaccacag |
| 6 | | tcaggaattcctacatttcattcatttttttttctctg |
| 7 | Production of chloramphenicol resistance disruption cassette | tgcaacaagagattcgcagccgcagtcttaaacaattggagga ataaggtatggagaaaaaaatcactgg |
| 8 | | ccattatacgacaaaccggcggctcggcgttagggctgatct cgactaagtcatcgcagtactgttgta |
| 9 | Screening of IF3 disruption and | caggaagttcgcttaacagg |
| 10 | transduction (PCR-product is synthesized in case of IF3 gene conservation only) | ggttagcgtgcttgtgc |
| 11 | Screening of IF3 disruption and | gacgtaaatgaagtgatcgagaag |
| 12 | transduction (PCR-products from IF3 gene and from disruption cassette are different in size) | ggttagcgtgcttgtgc |
| 13 | AIM23 cloning into pET32a | gactCATATGaatgcatcatctaccacaga |
| 14 | | ctagCTCGAGcatttcattcatttttttttctct |
| 15 | AIM23ΔNΔC cloning into pET32a | gactCATATGtggagcaccgggacaga |
| 16 | | ctagCTCGAGtggtttaacgtcctttggta |
| 17 | IF3 cloning into pET32a | atgcCATATGaaaggcggaaaacgagttc |
| 18 | | actgCTCGAGctgtttcttcttaggagcg |
| 19 | Screening of pET32a-based constructs | gctagttattgctcagcgg |
| 20 | | atgcgtccggcgtaga |

**Note:**
Restriction sites are in capital letters.

Western-blot was performed by standard protocol using the rabbit antibodies against 6-His-tagged recombinant Aim23p (produced on our order by Almabion).

## Construction of mutant *E. coli* strains

Genomic disruption of *infC* gene coding for IF3 was carried out in the *E. coli* strain MG1655. Cassette for *infC* genomic disruption containing the chloramphenicol resistance gene was prepared by PCR from pKD3 plasmid. Primers contained 5′-parts designed for the homologous recombination into the target genome site. The cassette was then delivered into *E. coli* cells by electroporation. These cells initially contained pKD46 plasmid encoding for recombinase, as well as pACDH plasmid containing *infC* gene. Clones where recombination took place were selected on chloramphenicol-containing medium and screened by PCR (*Thomason, Costantino & Court, 2007*).

For transferring the bacterial genetic material to phage P1, five ml of *E. coli* culture in logarhythmic growth phase was infected by 100 µl of phage suspension. The mixture was incubated at 37 °C for 3 h with shaking and centrifuged at $9,200 \times g$

for 10 min. The phage-containing upper fraction was taken and filtered through 0.45 µm filter.

For generation of the experimental *E. coli* strains, the MG1655 cells containing pBAD plasmid with cloned *infC* gene were transformed by the plasmids coding for IF3 or different variants of Aim23p. About two ml of ON cultures were pelleted and resuspended in one ml of 10 mM CaCl$_2$, five mM MgSO$_4$. Suspensions were 100 µl aliquoted, then half a volume of P1 lysate was added, and the mixtures were incubated at 37 °C for 30 min without shaking. Then one ml of lysogeny broth (LB) medium and 200 µl of sodium citrate were added followed by the incubation at 37 °C for 1 h with shaking. The cells were plated on the agar dishes with antibiotics, 0.02% arabinose and five mM sodium citrate. Screening of the clones was performed by PCR.

The growth curves of *E. coli* strains were registered in automatic mode using microplate reader Infinite M200 PRO (Tecan Instruments, Mannedorf, Switzerland).

## Ribosome purification and analysis

Ribosomes were isolated from *E. coli* strain MG1655 according to *Rivera, Maguire & Lake (2015)* with minor changes. Briefly, bacterial cells from one l culture with OD$_{600}$ ~0.6 were collected, lysed, ribosomes from clarified lysate were sedimentated through 10% sucrose cushion, and dissolved in minimal volume of 10 mM Tris–HCl pH 7.0; 60 mM KCl, 60 mM NH$_4$Cl, seven mM magnesium acetate, seven mM $\beta$-mercaptoethanol, 0.25 mM EDTA. Isolated ribosomes were stored at −80 °C. For dissociation reaction, approximately 24 pmoles (one unit of OD$_{260}$) of ribosomes were mixed with different amounts of recombinant Aim23p, IF3, or Aim23ΔNΔC in the above-indicated buffer. Mixtures were incubated at 37 °C for 30 min, and then applied on 15–40% continuous sucrose gradients prepared on the same buffer. Samples were centrifuged for 18 h at 100,000×*g*, and then fractionated from top to bottom (45 fractions each of 250 µl were taken). Absorbencies of all fractions at 260 nm were measured.

In case of cross-linking, between the incubation with proteins and loading on the sucrose gradient, formaldehyde was added to ribosome preparations up to 1%, and the mixtures were incubated for 30 min on ice.

## Molecular modeling

The homology model of the Aim23p complex with the *E. coli* 30S subunit was done with Modeller 9.17 (*Sali & Blundell, 1993*) (script may be found in Supplementary Information). For the building of this model, we have used the known structure of the bacterial 30S subunit complex with the cognate IF3 (*Pioletti et al., 2001*), as well as the sequence alignment of Aim23p with *E. coli* IF3 and other orthologs (*Atkinson et al., 2012*).

The folding of Aim23p N-terminal extension was done with the AbinitioFold protocol (*Bonneau et al., 2001*, *2002*) based of fragments obtained from the Robetta web server (*Kim, Chivian & Baker, 2004*). Simulation was stopped after 180,000 decoys were collected. The homology modeling with new conformation of N-terminal extension was done with

Modeller 9.17 (*Sali & Blundell, 1993*). Conformation of the C-terminal extension was equilibrated with FloppyTail protocol from Rosetta (*Kleiger et al., 2009*). The spatial structure visualization was done with the PyMOL Molecular Graphics System, version 2.0 (Schrödinger, LLC, New York, NY, USA).

## RESULTS

### Full-length Aim23p is undesirable for *E. coli* cells due to its terminal extensions

As it has been already mentioned in "Introduction," mammalian mtIF3 possesses some functional activity in *E. coli* cells (*Ayyub et al., 2018*). On the other hand, we have previously demonstrated that *E. coli* IF3 may partially rescue the growth defects of the yeast strain lacking Aim23p (*Kuzmenko et al., 2014*). Moreover, Aim23p was shown to bind the small subunit of bacterial ribosome in vitro (*Atkinson et al., 2012*).
Taken together, these findings have allowed us to hypothesize that Aim23p might be at least partially functional in *E. coli* cells as IF3.

To verify this hypothesis, we have constructed three plasmids for further delivery into *E. coli* cells, coding for either cognate IF3 (positive control), Aim23p, or Aim23p without its mitochondria-specific N- and C-terminal extensions (Aim23ΔNΔC). This last construct was designed in order to specifically check possible effects of Aim23p terminal extensions on bacterial translation: theoretically, these protein parts, being mitochondria-specific, might not be needed for protein biosynthesis in *E. coli*. Cloned genes of Aim23p and Aim23ΔNΔC did not contain sequences coding for mitochondrial targeting signal. Thereafter, we have disrupted *E. coli infC* gene coding for IF3. This gene contains the promoter for the expression of the downstream gene (*Wertheimer, Klotsky & Schwartz, 1988*), so we have removed only first 153 nucleotides of *infC* gene from the bacterial genome. Since IF3 is indispensable for bacteria, before disruption we have transformed *E. coli* with the plasmid bearing *infC* gene under control of glucose-repressible promoter. Finally, we have delivered the above-described plasmids into bacterial cells and disrupted the genomic copy of *infC* gene. The scheme of bacterial strains engineering is presented in Fig. 1A.

Then, we down-regulated the expression of *infC* gene in these strains with glucose and measured their growth rates. The resulting curves may be found in Fig. 1B. The strain bearing wild-type *infC* gene on the plasmid grows normally, with fast entering the logarithmic phase and reaching the plateau. The strain carrying empty vector shows no growth at first 10 h of incubation which is easily explained by the absence of *infC* gene. However, slow growth has been detected afterwards, probably as a result of glucose-repressed promoter leakage which, in turn, allows minimal amount of IF3 to be synthesized. If *E. coli* cells contain Aim23ΔNΔC, the corresponding strain's growth curve is almost identical to that of the strain containing an empty vector. This clearly indicated the impossibility of Aim23ΔNΔC to functionally substitute IF3 in bacterial cells. The most interesting case is definitely the bacterial strain bearing the full-size Aim23p. This strain, although reaching finally the level of the strain containing an

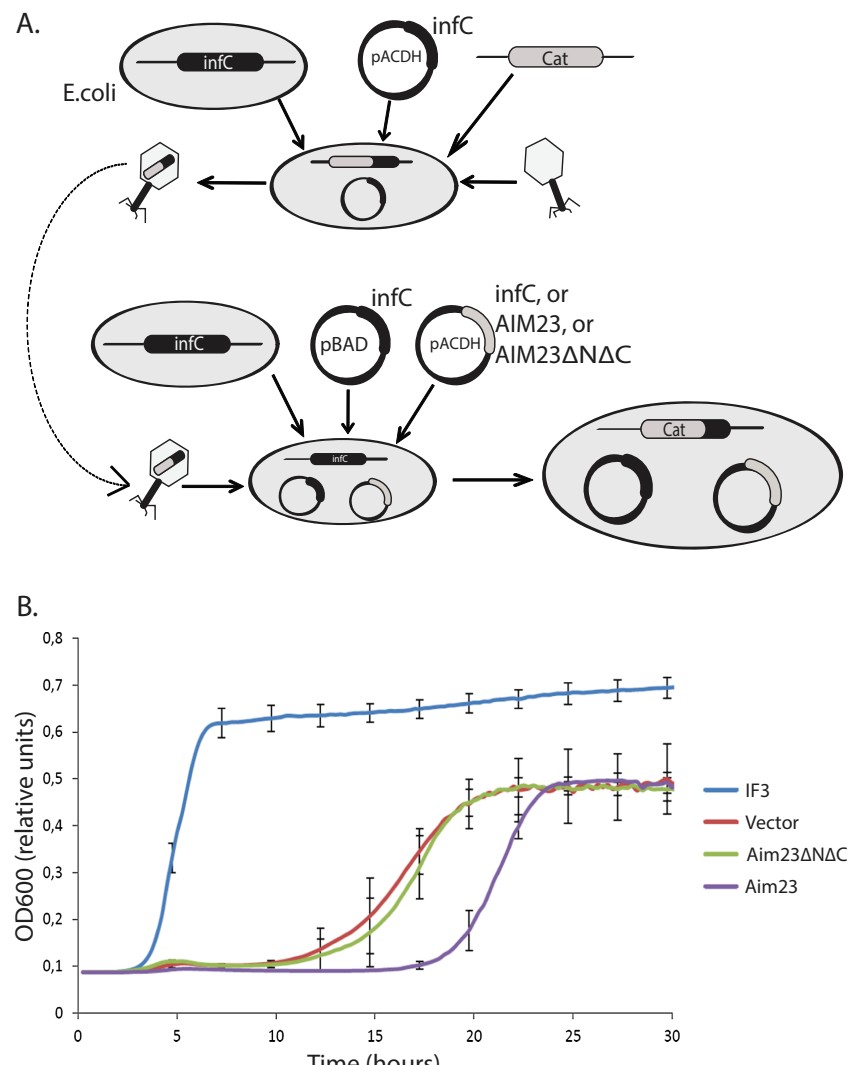

**Figure 1 Aim23p without terminal extensions is non-functional in *E. coli* cells while full-size Aim23p is even slightly toxic.** (A) Scheme of the mutant *E. coli* strains production. At the first stage, the *infC* gene coded for *E. coli* IF3 was cloned into pACDH vector. The resulting plasmid was delivered into *E. coli* cells following by the genomic disruption of *infC* by the chloramphenicol resistance gene (Cat). The *infC* gene on the plasmid made the resulting strain viable. Then, the cells were inoculating by P1 phage in order to capture the genomic DNA region containing the disrupted *infC* gene. The result of the first stage was the phage with the above-mentioned genomic DNA region. On the second stage, the *infC* gene was cloned into pBAD vector (under control of glucose-repressible promoter), and genes of Aim23p and Aim23ΔNΔC were cloned into pACDH vector. pBAD-*infC* and pACDH with one of the above-mentioned genes were then delivered into wild-type *E. coli* cells following by the inoculation by the phage from the first stage. This was resulted in the substitution of the wild-type *infC* genomic copy by the disrupted gene. As a result, a series of *E. coli* strains were generated with the following features: (1) genomic disruption of *infC*, (2) the presence of *infC* on the pBAD vector, (3) the presence of *infC* (positive control), Aim23p, or Aim23ΔNΔC on the pACDH vector. (B) Growth curves of the *E. coli* strains (indicated on the right) obtained as described in Fig. 3A. Bacteria were initially incubated without glucose, then the medium was changed to the glucose-containing one, and the optical density registration began. Each strain contained *infC* gene on the pBAD vector under control of glucose-repressible promoter. IF3: *infC* gene on the pACDH vector. Vector: empty pACDH. Aim23 and Aim23ΔNΔC: full-size and truncated AIM23 genes, respectively, on the pACDH vector.

empty vector, enters the logarithmic phase measurably slower than the latter. This means that the full-size Aim23p, but not Aim23ΔNΔC, negatively affects the viability of *E. coli* cells. It is rather possible that the terminal extensions of Aim23p may somehow interrupt the bacterial translation.

## Terminal extensions provide the ability of Aim23p to fix an unusual state of *E. coli* ribosomes

The above-described unusual effect of Aim23p in *E. coli* cells has led us to study the interaction of Aim23p with *E. coli* ribosomes in vitro. It is well known that adding cognate IF3 to purified bacterial ribosomes shifts the equilibrium of the ribosome dissociation reaction making the dissociated state preferable (*Gottleib, Davis & Thompson, 1975*). Based on this, we have purified ribosomes from *E. coli* cells, incubated them with the recombinant IF3 (positive control), or Aim23p, or Aim23ΔNΔC, fractionated the reactions by sucrose gradient centrifugation and analyzed the corresponding sedimentation profiles by measuring the optical densities of the fractions at 260 nm. The results of our experiment are presented in Fig. 2A. First of all, sedimentation profile of the ribosome sample with no proteins added was characterized by clear UV peaks of 30S and 50S subunits, as well as the whole 70S ribosomes, with the latter being most pronounced. Adding IF3, as expected, led to the complete dissociation of ribosomes into subunits (disappearance of the 70S peak and significant increase of the 30S and 50S peaks) while adding of Aim23ΔNΔC gave no effect on the sedimentation profile. This was also expected: in our in vivo experiments, this protein could not substitute for the cognate IF3 in *E. coli* cells (see Fig. 1). The profile of sedimentation has been curiously changed with adding the full-size Aim23p. This protein caused a fusion of 50S and 70S peaks with the formation of a single wide peak with maximum UV absorbance corresponded to approximately 60S sedimentation coefficient. At the same time, the 30S peak was increased, but to the less extent than in case of the full dissociation promoted by IF3. The most logical explanation of this phenomenon would be that Aim23p cannot promote the normal ribosome dissociation at concentrations used in the experiment (20:1 molar ratio in relation to the ribosomes concentration) but instead binds it and fixes this unusual state of ribosomes. The appearance of this "60S state" might be linked somehow to the Aim23p slight toxicity for *E. coli* cells observed by us (see Fig. 1B). It should be noted, finally, that such action of Aim23p on the bacterial ribosomes is definitely mediated by its mitochondria-specific terminal extensions.

The same ribosomal fractions were analyzed for presence of recombinant proteins by Western-blot hybridization. We used the homemade antibodies against 6-His-tagged recombinant Aim23p, and, luckily, they had a significant cross-reactivity with the 6-His-tag (data not shown). Thus, we were able to detect Aim23p, Aim23ΔNΔC, and IF3 since all recombinant proteins used in our experiments were 6-His-tagged. Results of this experiment can be found in Fig. 2B. *E. coli* IF3 was indeed detected only in 30S fractions while the Aim23p version without terminal extensions was not seen in either ribosomal fraction; instead, Aim23ΔNΔC was found in the very first fractions with no

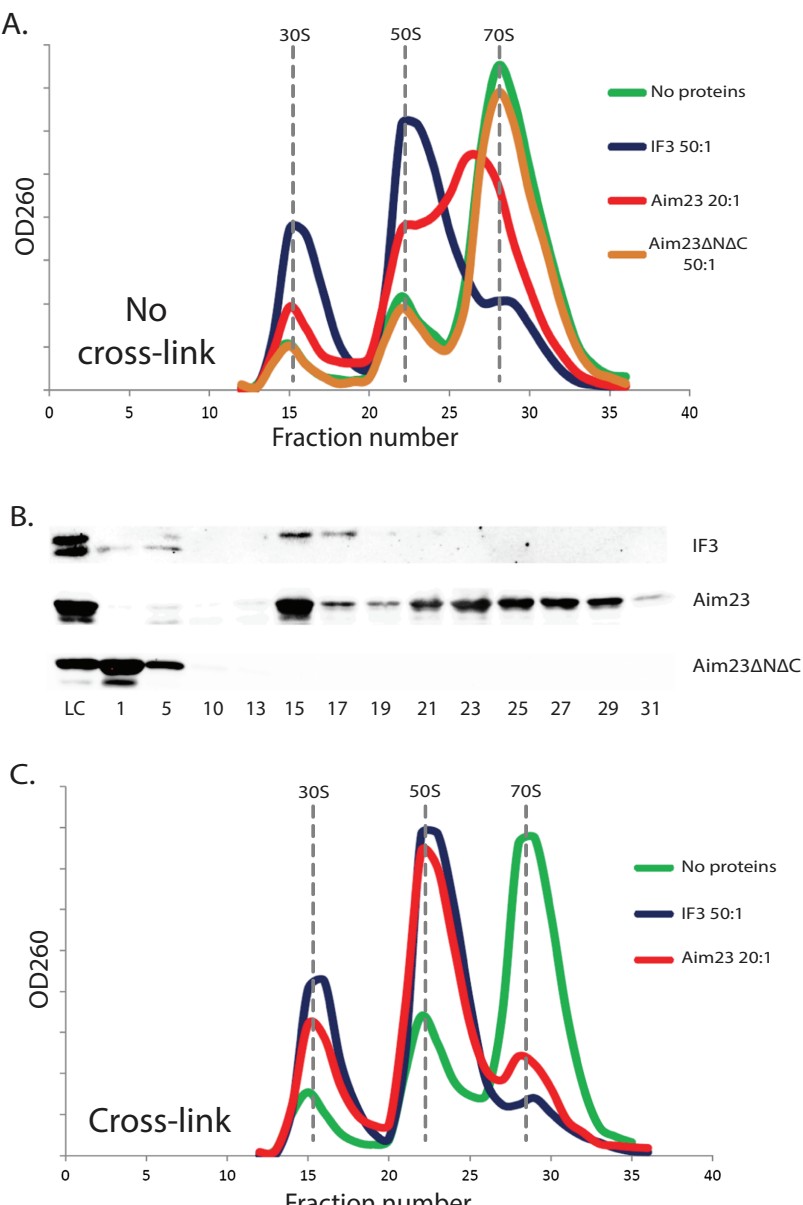

**Figure 2 The unusual effects of Aim23p on *E. coli* ribosomes in vitro.** (A and C) Ribosomes sedimentation profiles: optical densities at 260 nm (*Y*-axes of the graph) of different fractions of *E. coli* ribosomes which were pre-incubated with the indicated proteins, cross-linked with formaldehyde (C) or not (A), and sedimented in the sucrose gradient. On the *X*-axes: 20–25 sequential fractions, from bottom to top of the gradient. Molar ratios protein:ribosomes are indicated near each sedimentation profile. The peaks corresponded to the ribosomes and their free subunits are marked with the vertical dotted lines. (B) Western-blot hybridization of different fractions of *E. coli* ribosomes pre-incubated with different proteins. The same ribosomes samples as in (A) were analyzed for this experiment. We used the antibodies against recombinant Aim23p with the significant cross-reactivity to the 6-His-tag which allowed us to detect both Aim23p and IF3. Numbers of fractions analyzed are indicated below. Abbreviation: LC, loading control.               

ribosomes. This explains the impossibility of Aim23ΔNΔC to promote dissociation of bacterial ribosomes: this protein binds neither 70S ribosomes nor their separate subunits. Interestingly, full-size Aim23p behaves in all the contrary way compared to

its version lacking terminal extensions. This protein is detectable in nearly all fractions containing ribosomes, either assembled or in the form of subunits. Maximum amount of Aim23p is bound to free 30S subunits, and there is almost equal distribution of the protein between fractions corresponding to 50S and "60S" peaks. Such non-canonical manner of binding ribosomes might be one of the reasons why Aim23p promotes formation of their "60S" state.

### "60S state" of *E. coli* ribosomes is not a stable structure

The observed peak at 60S zone of the sedimentation profile may be explained in at least two ways. The first explanation is that Aim23p binds *E. coli* ribosomes and causes changes in their structure so that their sedimentation coefficient decreases to 60S. An alternative explanation is that, upon Aim23p binding, the ribosomal subunits become more flexible relative to one another. This, in turn, allows their reciprocal movements without full dissociation. In this case, the observed 60S peak might reflect the changed dynamic equilibrium between associated and dissociated states of the ribosome, rather than formation of a stable structure. In order to distinguish between these two possibilities, we have repeated our ribosomes fractionation experiment with additional cross-linking step (with the help of formaldehyde) after incubation with proteins. We did not use Aim23ΔNΔC here since this protein is unable to bind *E. coli* ribosomes (Figs. 2A and 2B). The results of cross-linking experiment are presented in Fig. 2C. After the formaldehyde treatment, the sedimentation profiles of ribosomes incubated with Aim23p or IF3 were almost identical, and there was no 60S peak of the Aim23p-bound ribosomes. If 60S peak would reflect the formation of a stable ribosomal structure, this structure would be fixed by cross-linking. Thus, our results clearly indicate that the observation of 60S peak is the consequence of some dynamic processes caused by Aim23p binding. One may speculate that these processes are the very first stages of 70S ribosomes dissociation which cannot continue normally due to the unusual manner of Aim23p interaction with the ribosomes.

After obtaining these intriguing results, we decided to analyze the dose-dependency of the Aim23p effect on *E. coli* ribosomes. The resulting profiles of ribosomes sedimentation after adding Aim23p at different concentrations are presented at Fig. 3A. If Aim23p concentration is 2.5 times more than in previously described experiment (i.e., 50:1 molar ratio in relation to the ribosomes concentration), then the peak of "60S state" is almost not observed. Instead, one can see a normal 50S peak which is slightly moved toward the increase of the sedimentation coefficient. At the same time, the 30S peak is meaningfully increased relative to the situation when the "60S state" is clearly observed. When Aim23p concentration is increased twice more (up to 100:1 molar ratio in relation to the ribosomes concentration), the resulting profile is identical to that in case of IF3 adding to ribosomes. One can hypothesize that these results reflect the consecutive stages of 70S ribosomes dissociation through "60S state" to free 30S and 50S subunits.

### Aim23p and *E. coli* IF3 act jointly to dissociate bacterial ribosomes in vitro

The discovered "60S state" of bacterial ribosomes might be the result of the decreased ribosome stability. In other words, the equilibrium of the dissociation reaction in this case

A.

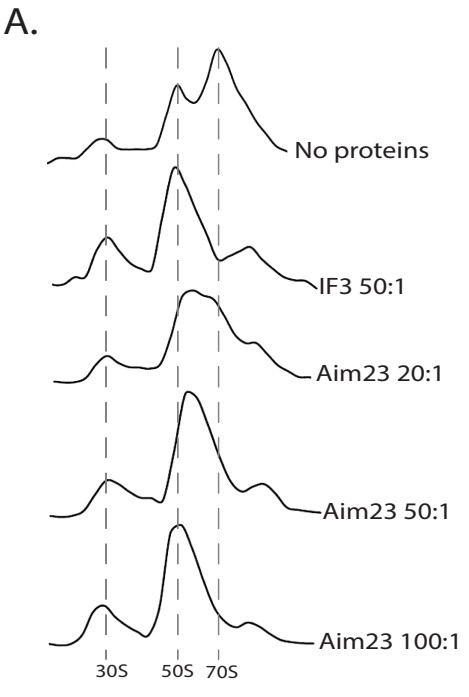

B.

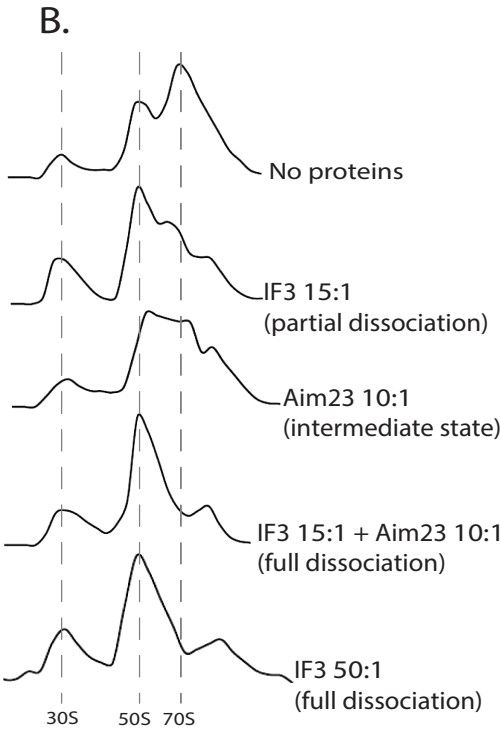

C.

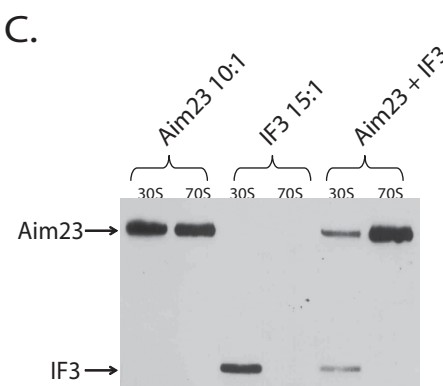

**Figure 3 Aim23p is able to dissociate *E. coli* ribosomes either in large concentrations, or together with *E. coli* IF3.** (A and B) Ribosomes sedimentation profiles: optical densities at 260 nm (*Y*-axes of the each graph) of different fractions of *E. coli* ribosomes which were pre-incubated with the indicated proteins and sedimented in the sucrose gradient. On the *X*-axes: 20–25 sequential fractions, from bottom to top of the gradient. Molar ratios protein:ribosomes are indicated near each sedimentation profile. The peaks corresponded to the ribosomes and their free subunits are marked with the vertical dotted lines. (C) Western-blot hybridization of different fractions of *E. coli* ribosomes which were pre-incubated with the indicated proteins and sedimented in the sucrose gradient. In each case, the mixture of two fractions composing the peaks of 30S or 70S was analyzed (indicated on the top). Two fractions composing the corresponding peaks in Fig. 3B were combined and loaded on PAAG. We used the antibodies against recombinant Aim23p with the significant cross-reactivity to the 6-His-tag which allowed us to detect both Aim23p and IF3 (indicated by arrows) in the single analysis.

may be slightly shifted toward free subunits without full dissociation. This, in turn, means that such state of the ribosome should be subjected to dissociation easier than the normal 70S state. In order to check this hypothesis, we performed in vitro dissociation experiments with Aim23p and IF3 being simultaneously added to ribosomes. While Aim23p was added in concentration sufficient for "60S state" fixation (10:1 molar ratio in relation to the ribosomes concentration; see Fig. 3B), the amount of IF3 used was not enough for full ribosome dissociation (15:1 molar ratio in relation to the ribosomes concentration). If both proteins were presented in the reaction together (each at the same concentration as alone), the complete dissociation was detected. This phenomenon could be explained as follows. If "60S state" appears as a result of Aim23p action, the minimal amount of the free 30S subunits is immediately formed (this was also seen in our previous experiments, see Figs. 2A and 3A). Adding a little amount of the cognate IF3 leads to the fixation of the 30S subunits in their free state and further shifts the reaction equilibrium toward the dissociated state of the ribosome. Thus, Aim23p and IF3 may act jointly to promote the dissociation of the bacterial ribosomes. To verify this, we performed a Western-blot analysis of the ribosomes fractions corresponding to the free subunits and to the whole ribosomes in presence of Aim23p, or IF3, or both proteins together. The results are presented at Fig. 3C. IF3 in this experiment has been found to bind exclusively free 30S subunits but not 70S ribosomes, exactly as expected. Aim23p, however, is detected both in free 30S subunits and in the 70S ribosomes fractions (which fits well to our results presented in Fig. 2B), and this does not qualitatively depend on presence or absence of IF3 in the reaction. This explains well the joint action of these two proteins resulting in the ribosomes dissociation which cannot be achieved when using Aim23p and IF3 at the same concentrations separately.

## Molecular modeling points on the importance of Aim23p terminal extensions for protein interaction with *E. coli* ribosomes

A very interesting question rises from the above-described results: in which manner does Aim23p interact with bacterial ribosome and what is the role of its terminal extensions in such an interaction? To shed light on this problem, we have performed molecular modeling.

Previously (*Atkinson et al., 2012*), sequence alignment of Aim23p with *E. coli* IF3 and other orthologs has been done. On the base of this data, as well as the known structure of 30S complex with the cognate IF3 (*Pioletti et al., 2001*), we have built the homology model of Aim23p complex with *E. coli* 30S ribosomal subunit with the help of Modeller 9.18 (script may be found in Supplementary Information). In the resulting model, Aim23p eventually has a long and extended N-terminal tail (Fig. S2). The size of this tail was comparable with the size of the 30S subunit, and the model could not provide valuable information about N-terminal extension function. We have suggested that the N-terminal extension is somehow structured and have built the corresponding model with the Rosetta AbInitio protocol. From 18,398 decoys of N-terminal extension, the top 10 had an alpha-helical structure with RMSD less than 10 Å. This observation reflects the fact that the N-terminal extension does not possess a certain spatial structure but probably

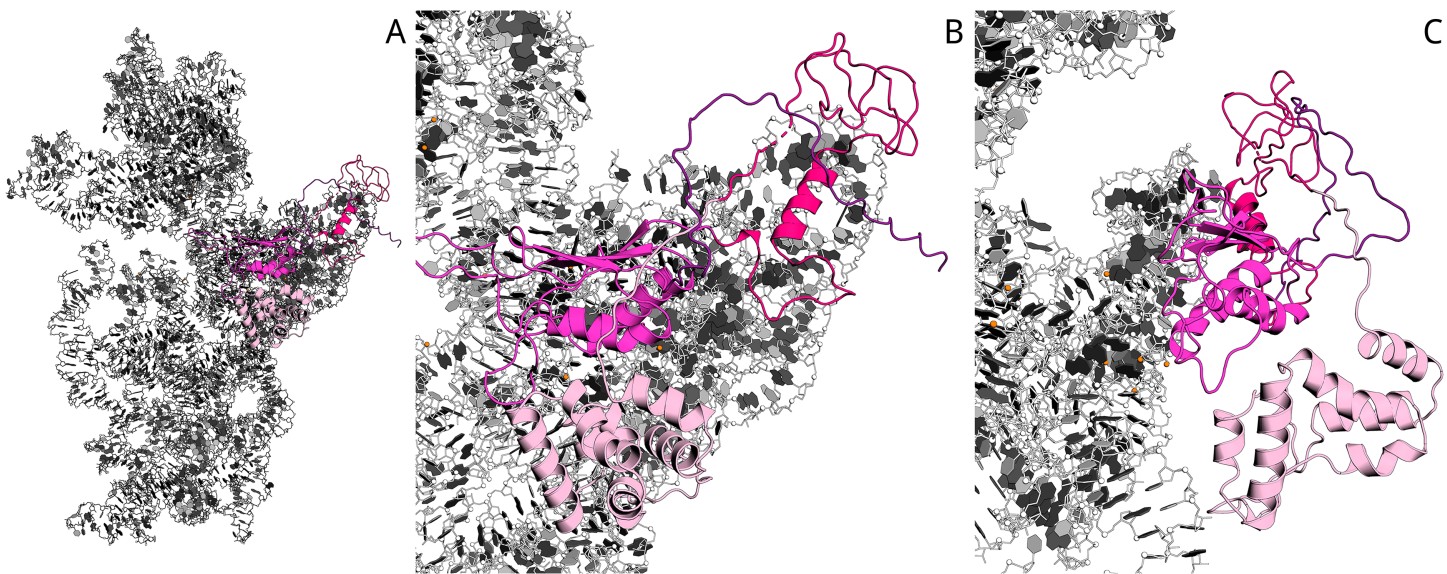

**Figure 4 Model of Aim23p interactions with *E. coli* 16S RNA.** N-terminal extension is in light-pink, N-terminal domain is in hot-pink, C-terminal domain is in magenta and C-terminal extension is in deep-purple. 16S RNA is in black and white. (A) Overview of Aim23p location on 16S RNA. (B) Close-up view in same orientation. (C) Close-up view with counterclockwise rotation around vertical axis displaying proximity of N-terminal extension, C-terminal domain and 16S RNA.

has a mobile helical packaging. The N-terminal extension model with the best Rosetta score was used to rebuild the new homology model of 16S RNA and the Aim23p complex. The resulting model surprisingly revealed a strong interaction of the N-terminal extension with the C-terminal domain of the Aim23p core part and with the long 3′ terminal helix of 16S RNA. Additional distance restraints between centers of mass from 15 to 40 Å were applied to sample distance between Aim23p's N-terminal extension and C-terminal domain. As a result, the top models have confirmed the interaction of the N-terminal extension with the C-terminal domain, while interaction with 16S RNA does not look favorable. The best models of packed C-terminal extension showed interaction with the N-terminal domain. Thus, in silico modeling points to the possible mode of Aim23p interaction with 30S subunit where the terminal extensions of the protein "press down" the core Aim23p part to the ribosome. This may be the reason for the importance of the Aim23p terminal extensions for binding bacterial ribosomes (see Fig. 2). The summary of molecular modeling is presented in Fig. 4.

## DISCUSSION

We have demonstrated previously that *E. coli* IF3 fused with the mitochondrial targeting sequence of Aim23p may complement to a minimal extent the absence of the AIM23 gene in yeast (*Kuzmenko et al., 2014*) which is strong evidence for Aim23p being a bona fide IF3 in mitochondria. This finding is not surprising taking into account that bacterial enzymes may often functionally substitute for their mitochondrial orthologs in the organelles. This, for example, has been demonstrated for several aminoacyl-tRNA synthetases (*Edwards & Schimmel, 1987*; *Chiu, Chang & Wang, 2009*) and for the proteins involved in Fe-S clusters formation (*Kispal et al., 1999*). In this work, we have

performed a "reverse" experiment and investigated if Aim23p is able to substitute for cognate IF3 in *E. coli* cells. The cases of successful complementation of bacterial proteins by their mitochondrial orthologs have been described remarkably rarer than the opposite situations. However, mammalian mitochondrial initiation factor 2 has been shown to function in *E. coli* cells instead or two cognate factors at once, namely IF1 and IF2 (*Gaur et al., 2008*). Most probably, this is due to the short insertion domain of mammalian mtIF2 that is believed to execute the function of IF1 in mitochondria. Moreover, in a recent work it has been demonstrated that mammalian mtIF3, although not being able to fully substitute for IF3 in *E. coli*, exhibits some functional activity in bacterial cells (*Ayyub et al., 2018*). Speaking about Aim23p, this protein, as we have discovered in the present study, does not work as an initiation factor in *E. coli*, independently of presence or absence of the terminal extensions. We used an experimental system where cognate IF3 gene was disrupted in the bacterial genome but was presented on the plasmid under the control of glucose-repressible promoter. Such promoters are well known to leak if the amount of glucose is low. In our case, this ensures synthesis of minimal amount of IF3 and weak growth of the bacterial culture after a dozen of hours of incubation, when the main portion of glucose becomes utilized by bacterial cells (Fig. 1B). Surprisingly, this weak growth is even slower in presence of full-size Aim23p when comparing to Aim23p without terminal extensions. This means that these regions of Aim23p even make this protein slightly toxic for bacterial cells. Interestingly, mammalian mtIF3 behaves quite differently in *E. coli*. Full-size factor does not markedly affect the *E. coli* growth rate while deletion of the N-terminal extension leads to the severe growth impairment (*Ayyub et al., 2018*). However, to our opinion, these results should not be directly compared with the data presented in this work. The main reason for this is the difference in the experimental systems. Ayyub et al. used the mutant strain in which IF3 was devoid of first 55 amino acids and was synthesized in normal quantities. Earlier, the same authors have shown that this truncated version of IF3 is enough for *E. coli* survival and can perform all main functions of the factor (*Ayyub, Dobriyal & Varshney, 2017*). This means that the action of any mtIF3 version in such cells is somewhat additional to the action of the cognate factor. On the contrary, our *E. coli* cells contained minimal amount of wild-type IF3 synthesized from repressed but leaking promoter, and the quantity of Aim23p encoded in the plasmid was much higher. In this case, the heterologous factor influence on the bacterial cells might be stronger than that discovered by Ayyub et al.

The negative influence of Aim23p on *E. coli* cells, most probably, might realize via its interaction with bacterial ribosomes. This is exactly what we have demonstrated in the present work. In certain concentration range, Aim23p promotes the formation of a very unusual state of *E. coli* ribosomes in vitro. Our results presented in Figs. 2 and 3 indicate that this state is characterized by the partial fusion of 70S and 50S peaks. The maximum of absorbance at 260 nm in this case approximately corresponds to the 60S sedimentation coefficient. We propose to call it "60S state." To our current knowledge, such ribosome state has never been detected in vivo. However, it was described in in vitro experiments (*Morimoto, 1969*), notably at approximately the same magnesium

concentrations as we used in our work (10 vs 7 mM, respectively). Morimoto used a term "60S component" and postulated that this was a new stable intermediate of the subunits association reaction and that this intermediate was just "swollen 70S" (*Morimoto, 1969*). Further investigations, however, have demonstrated that the sedimentation coefficient of this "swollen 70S" depends on the centrifugation speed and on the initial 70S concentration (*Spirin, 1971*). This indicates that discussed ~60S zone on the sedimentation pattern is the consequence of the dynamic equilibrium of dissociation-association reaction rather that the stationary ribosomal structure. In our work, we came to the same conclusion when treating "60S state" with formaldehyde (Fig. 2C). Such treatment led to complete disappearance of the corresponding peak showing that the 60S state of the ribosomes cannot be regarded as a stable structure.

In this work, the 60S peak on the ribosomes sedimentation profile has been for the very first time detected after adding a protein to the ribosomes. Aim23p possesses this activity due to its terminal extensions since we have not seen any changes in the ribosome sedimentation profile when adding Aim23ΔNΔC (Fig. 2A). In order to elucidate the role of terminal extensions, we have performed molecular modeling of Aim23p complex with 30S. According to its results, the direct interaction between terminal extensions of Aim23p (especially N-terminal one) and the core protein part might take place. This probably makes Aim23p "fixed" on the small subunit (see Fig. 4). Interestingly, the similar effect of mammalian mtIF3 has been described with regard to human mitochondrial ribosomes dissociation in vitro (*Haque, Grasso & Spremulli, 2008*). Full-length mtIF3 promotes normal dissociation while using its truncated version without terminal extensions causes the partial fusion of the 39S (large subunit) and 55S (whole mitoribosome) peaks. This clearly indicates some abnormal dissociation.

In the present work, the "60S state" has been demonstrated to dissociate by increased amount of Aim23p (Fig. 3A), or by a small amount of *E. coli* IF3 (Fig. 3B). The slight toxicity of Aim23p for *E. coli* cells (Fig. 1B) may be also explained by fixing the "60S state" which dissociates poorly in presence of marginal IF3 quantities synthesized from leaking promoter. At the same time, the presence of even a large amount of Aim23p in *E. coli* cells together with the physiological amount of the cognate IF3 has no effect on bacterial viability, as we could see when purifying recombinant Aim23p from wild-type *E. coli* cells (data not shown). Even if the "60S state" is fixed in such conditions, it might rapidly dissociate to the subunits with the help of IF3 since this state might dissociate easier than 70S ribosomes (see Fig. 3B). This may be also the case in the work of *Ayyub et al. (2018)*: having sufficient amounts of cognate *E. coli* IF3 allows bacterial ribosomes to keep the dissociated state in vivo properly regardless on the mammalian mtIF3 presence, and this could explain the almost normal growth of the corresponding bacterial strains.

The question if "60S state" exists in wild-type bacterial cells is of high interest. The answer "no" seems to be obvious, as bacterial IF3 is well-known to bind only free 30S subunits. This was also seen in the present work (Figs. 2B and 3C). At the same time, IF3 must bind 70S ribosomes, or at least stay bound to 30S when 70S is already formed,

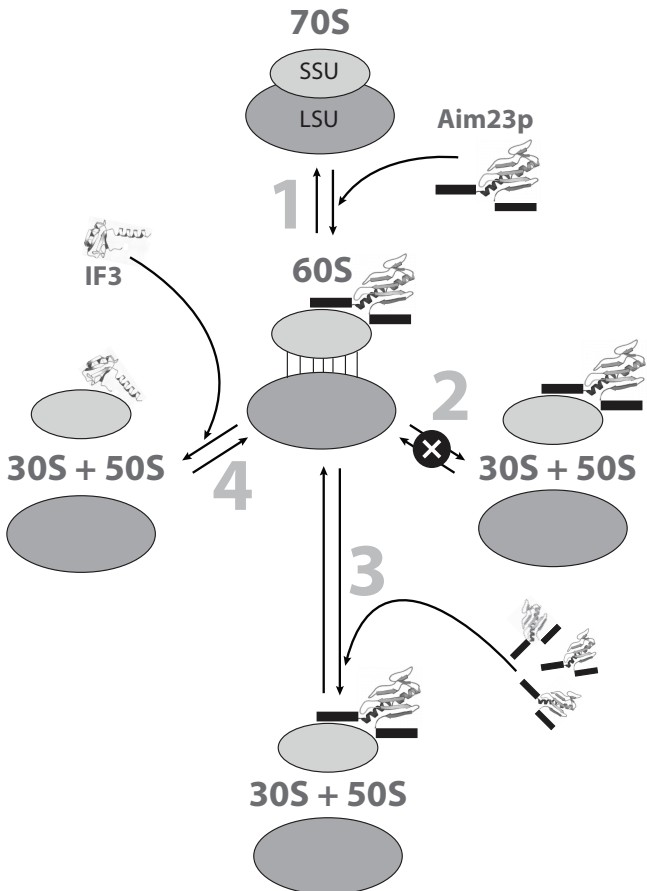

**Figure 5 The hypothetic scheme of the formation and dissociation of *E. coli* ribosomes intermediate state in vitro.** 1. Initially, the small (SSU) and large (LSU) subunits of the ribosome are associated one to another (70S). Adding Aim23p (the terminal extensions are represented by black boxes) changes the ribosome conformation, making the subunits more flexible relative to one another and allowing their reciprocal movements without full dissociation (60S). 2. This intermediate dissociation state cannot spontaneously dissociate to the subunits in the presence of Aim23p. 3. Adding more Aim23p, however, shifts the dissociation reaction equilibrium, which results in appearance of the free SSU and LSU (30S + 50S). 4. Full dissociation of the intermediate can also be reached by adding *E. coli* IF3 in an amount insufficient for dissociation of 70S ribosomes.

to fix any dissociation intermediate. The impossibility of this binding, however, is not dogmatic. In the structural study, IF3 was found as a part of the fully assembled bacterial initiator complex, together with 70S ribosomes (*Allen et al., 2005*). The authors propose that IF3 does bind the free 30S subunit initially and then remains bound to 70S ribosomes for a short time after subunits association. Moreover, in a recent study binding of IF3 with 70S ribosome was confirmed by Förster resonance energy transfer (FRET) microscopy experiments, and an alternative binding site of IF3 was identified on 50S subunit (*Goyal, Belardinelli & Rodnina, 2017*). The subunits association in presence of IF3 might be realized via some intermediate states relative to the "60S state" detected in the present work.

The possible mechanisms of the *E. coli* ribosomes 60S intermediate state formation and dissociation are summarized in Fig. 5.

## CONCLUSIONS

The main result of this work is the detection of a state of *E. coli* ribosomes ("60S state") which is formed as a result of interaction with *S. cerevisiae* mitochondrial translation IF3, Aim23p. We also demonstrate that Aim23p and cognate *E. coli* IF3 actions on bacterial ribosome are of different modes and that these two proteins may bind it jointly. We show that the key players in the game of Aim23p binding to *E. coli* ribosomes are the protein's mitochondria-specific terminal extensions that, according to the molecular modeling results, might nestle the core part of Aim23p to a ribosomal small subunit. Thus, the binding efficiency increases. Our results provide a basis for future structural studies of "60S state" which, in turn, will elucidate the fine mechanisms of bacterial ribosome dissociation/association.

## ACKNOWLEDGEMENTS

We are grateful to Gemma Atkinson (Umea University, Sweden, and Tartu University, Estonia) for the in silico prediction of Aim23p terminal extensions. We also thank Konstantin Khodosevich (Copenhagen University, Denmark), Vasili Hauryliuk (Umea University, Sweden, and Tartu University, Estonia), Ivan Tarassov (Strasbourg University, France), Stanislav Kozlovsky and Alexey Kharitonov (Moscow University, Russia) for providing strains and chemicals. We appreciate the improvement of the figures quality by Alexey Fedyakov (Moscow University, Russia). Our special thanks are to Sergey Dmitriev (Moscow University, Russia) and to Vyacheslav Kolb and his lab members (Institute of Protein Research, Russia) for helpful discussions. The technical help of our students Maria Klimontova, Valeria Zinina, Anna Mirnaya, Anastasia Kapusta, and Margarita Chudenkova is greatly appreciated. Molecular modeling was carried out using the equipment of the shared research facilities of HPC computing resources at Lomonosov Moscow State University.

### Funding

This work was supported by the Russian Science Foundation (Grant 17-14-01005 for experimental work and Grant 14-50-00029 for molecular modeling). The funders had no role in study design, data collection and analysis, decision to publish, or preparation of the manuscript.

### Grant Disclosures

The following grant information was disclosed by the authors:
Russian Science Foundation (Grant 17-14-01005 for experimental work and Grant 14-50-00029 for molecular modeling).

### Competing Interests

The authors declare that they have no competing interests.

### Author Contributions

- Sergey Levitskii conceived and designed the experiments, performed the experiments, analyzed the data, prepared figures and/or tables, approved the final draft.

- Ksenia Derbikova performed the experiments, analyzed the data, prepared figures and/or tables, approved the final draft.
- Maria V. Baleva performed the experiments, analyzed the data, approved the final draft.
- Anton Kuzmenko conceived and designed the experiments, performed the experiments, analyzed the data, approved the final draft.
- Andrey V. Golovin conceived and designed the experiments, performed the experiments, analyzed the data, prepared figures and/or tables, approved the final draft.
- Ivan Chicherin performed the experiments, approved the final draft.
- Igor A. Krasheninnikov analyzed the data, prepared figures and/or tables, approved the final draft.
- Piotr Kamenski conceived and designed the experiments, analyzed the data, prepared figures and/or tables, approved the final draft.

## Data Availability

The raw data are provided in the Supplemental Files.

## Supplemental Information

Supplemental information for this article can be found online at http://dx.doi.org/10.7717/peerj.5620#supplemental-information.

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
