# Peer review of "S dynamic state of bacterial ribosome is fixed by yeast mitochondrial initiation factor 3"

_PeerJ, doi:10.7717/peerj.5620_

## Round 0.1 · original submission · Major Revisions

Both reviewers have serious concerns about the experimental design, a lack of controls and validations by orthogonal experiments. As it stands the manuscript is not acceptable. To make the manuscript acceptable you need to address the experimental issues including controls and validation by orthogonal experiments as outlined by reviewer 2 In addition you should tone down your discussion to give a more appropriate and balanced interpretation of the results.

Reviewer 1 ·

Basic reporting

The language needs substantial improvement, the citations are sufficient.

Experimental design

The rational for the experiments are not well-elaborated and the physiological significance is unclear. The experimental design is poor and lacks obvious controls.

Validity of the findings

The manuscript revolves around two sets of experiments that are not vaildated by additional, well-approachable experiments to determine kinetics of subunit dissociation. Data are therefore not robust and do not support the conclusion or the proposed model.

Additional comments

More work will be required to elaborate the findings of this manuscript. Standard methodology can be applied.

Reviewer 2 ·

Basic reporting

The paper is written with clarity with appropriate structure and context and the article meets the requirements for providing raw data for PeerJ.
The authors provide a series of experiments including bacterial growth curves, ribosome profiles augmented with recombinant proteins and molecular modelling. For the most part the figures are clear and easy to follow.

Experimental design

The original aims are clear - the authors wished to see if a yeast mitochondrial factor can complement for a bacterial initiation factor by analogy to a related study where this was done with a mammalian factor. The study runs a risk of being phenomenological, as it is not clear that the entity they describe has any relation to physiological complexes that might actually form.

Issues related to technical rigour and description of methods are as follows:

1. Growth curves for bacterial cells do not appear to have error bars or they might not be very large. The raw data suggest that a number of technical replicates were made but it is not clear how this relates to the data in the graph – were mean values and standard deviations calculated? And plotted?

2. The western blot in Fig. 2B is used as evidence that Aim23deltaNdeltaC cannot bind to the ribosomal subunits. These data are incomplete. The authors need to show the input samples as well as the samples taken from the gradient, particularly as they only sample two peaks. How are we to know that the researcher who carried out the experiment didn’t simply forget to add the reagent to those samples?

Validity of the findings

On the basis of their data, the authors conclude conclude that the yeast mitochondrial initiation factor is stabilizing a new “60S” form of the ribosome that is locked in pre-dissociated state. There are some issues with this interpretation, mainly because it relies on just a few observations from one kind of experiment and there are no orthogonal methods to support these data. It is important that the authors tone down their claims as the language used is far too strong.

1. It is not clear what the molecular modelling brings to the study as the proposed model has not been validated experimentally. The authors should avoid discussing the model as if it has high precision e.g.

Line 357-9 ‘This has been confirmed by the molecular modelling … terminal extensions of Aim 23p … have been shown to interact directly with the core protein part which probably makes Aim32P “fixed” on the small subunit.’

2. The observation that a 60S particle is formed is based on the ribosome profiling curves presented in figure 2. The apparent 60S peak might really be the sum of the 50S and 70S curves, suggesting that it is not a new entity that is being formed, but rather Aim23p is inefficient at subunit dissociation. For a 60S particle to form it suggests that the ribosome would become more compact compared to the 70S to have a smaller hydrodynamic radius. It is hard to imagine how this would occur in the context of all of the structural data available for ribosome particles. Furthermore, if Aim23p were stabilising a dissociation complex, one might expect that it would act as a dominant negative in cells, yet this is contrary to what the authors observe [line 371-2 ‘… the presence of even large amount of Aim23p in E. coli cells with physiological amount of the cognate IF3 has no effect on bacterial viability…’]. One would also expect to see more 60S at higher concentrations of Aim23p and again this is contrary to what the authors observe (Fig 2B).

Unless there are orthogonal data to support the existence of the 60S particle directly, it is not appropriate to state that:

Line 209 ‘…we have to hypothesize that Aim23p somehow affects the whole ribosome structure…’
Line 248-9 ‘The only possible explanation of this phenomenon is that “60S state” is indeed subjected to dissociation easier than the normal 70S state.’
Line 354 ‘…the 60S intermediate dissociation state has been for the very first time fixed by adding a protein to the ribosomes…’

These statements should be toned down to give a more nuanced interpretation.

Additional comments

no comment

---

## Round 0.2 · accepted · Accept

You adressed the concerns apropriately. I think you made a valid observation, which is worth sharing with the public. However, to me, the importance and nature of the dynamic 60S-state remains unclear.

#